# A Plant-Derived Antifungal Agent, Poacic Acid, Inhibits Germination and Tube Growth of Lily Pollen

**DOI:** 10.3390/plants14193093

**Published:** 2025-10-07

**Authors:** Nanami Kobayashi, Yoshikazu Ohya, Yasuko Hayashi, Shuh-ichi Nishikawa

**Affiliations:** 1Graduate School of Science and Technology, Niigata University, Niigata 950-2181, Japan; snchehon117@gmail.com; 2Department of Integrated Biosciences, Graduate School of Frontier Sciences, The University of Tokyo, Kashiwa 277-8562, Japan; ohya@vos.nagaokaut.ac.jp; 3Department of Science and Technology Innovation, Nagaoka University of Technology, Nagaoka 940-2188, Japan; 4Faculty of Science, Niigata University, Niigata 950-2181, Japan; yhayashi@env.sc.niigata-u.ac.jp

**Keywords:** poacic acid, pollen, β-1,3-glucan, callose, membrane traffic, actin cytoskeleton

## Abstract

Poacic acid is a novel natural antifungal agent. It inhibits the growth of fungal cells, including budding yeast *Saccharomyces cerevisiae*, by inhibiting the synthesis of β-1,3-glucan, which is a major component of the fungal cell wall. Although poacic acid is expected to be a candidate pesticide owing to its antifungal activity, its effects on plant cells have not been investigated. In this study, we analyzed the effects of poacic acid on lily (*Lilium longiflorum*) pollen. Poacic acid inhibited lily pollen germination and tube growth at concentrations lower than those that inhibited budding yeast growth. While poacic acid did not inhibit callose (β-1,3-glucan) synthesis in pollen tubes, it inhibited membrane traffic, including endocytosis and secretion, and the organization of the actin cytoskeleton within pollen tubes. Since these processes have been shown to play essential roles in pollen tube growth, our study indicates that poacic acid affects lily pollen tube growth differently than it affects budding yeast. Poacic acid also inhibited the growth of *Arabidopsis thaliana* pollen tubes and root.

## 1. Introduction

Poacic acid is a novel natural antifungal agent found in lignocellulosic hydrolysates from corn stover [1,2]. Poacic acid inhibits the growth of *Saccharomyces cerevisiae* by inhibiting the synthesis of β-1,3-glucan, a major and essential component of the yeast cell wall. The inhibitory mechanism of β-1,3-glucan synthesis by poacic acid differs from that of candin family antifungal agents, including caspofungin. In yeast, the synthesis of β-1,3-glucan is mediated by glucan synthase [3]. Caspofungin binds to Fks1, the catalytic subunit of glucan synthase, and reduces its activity of Fks1 by changing its conformation of Fks1 [4,5]. In contrast, poacic acid inhibits β-1,3-glucan synthesis by binding to β-1,3-glucan [1,2]. Although both compounds inhibit β-1,3-glucan synthesis, poacic acid induces distinct transcriptional profiles compared to caspofungin [2]. Consistent with these differences, the sensitivity of yeast species to poacic acid does not correlate with their sensitivity to caspofungin [6].

Poacic acid inhibits the growth of several fungal species, including the phytopathogenic fungi *Sclerotinia sclerotiorum* and *Alternaria solani*, as well as *S. cerevisiae* [1]. In addition to glucan synthesis, poacic acid is implicated in cell wall remodeling [2]. Given its activity against phytopathogenic fungi, poacic acid is expected to be useful as a pesticide. To explore its practical potential as a pesticide, the effects of poacic acid on plant cells must be studied. However, the effects of poacic acid on plant cells have not yet been investigated.

Pollen, as a haploid gametophytic generation, is important for the sexual reproduction of flowering plants. Pollen germination and rapid tube growth require various essential cellular processes, including membrane trafficking and cytoplasmic streaming [7,8]. Inhibition of these processes results in the arrest of pollen tube growth or the development of aberrant tube morphologies. Owing to these easily observable changes in growth and morphological phenotypes, pollen tubes have been used to evaluate the effects of compounds on plant cells and screen for chemicals that affect the activities of plant cells [9,10,11,12,13,14]. This led us to use pollen tubes to analyze the effects of poacic acid on plant cells.

Pollen tubes have another characteristic that can be used to analyze the effect of poacic acid, which is the active synthesis of β-1,3-glucan, known as callose, in the pollen tube cell wall. Callose is synthesized transiently in various plant tissues at different stages or upon pathogen infection and plays a role in regulating plant development and physiological responses [15]. Callose is a major cell wall component of pollen tubes, and callose plugs are periodically deposited to separate the growing tip from the evacuated portions of the tube [16]. Plant callose synthases share homology with fungal FKS proteins [17].

In this study, we analyzed the effects of poacic acid on lily (*Lilium longiflorum*) pollen. Poacic acid inhibited lily pollen germination and tube growth at concentrations lower than those that inhibited budding yeast growth. Poacic acid did not inhibit callose synthesis in the pollen tubes. We found that poacic acid was incorporated into pollen tubes and inhibited endocytosis and secretion, as well as perturbing the organization of the actin cytoskeleton in the pollen tubes. These results indicate that poacic acid affects lily pollen tube growth differently than it affects budding yeast.

## 2. Results

### 2.1. Poacic Acid Inhibits Lily Pollen Germination and Pollen Tube Growth

We found that poacic acid inhibited the germination of lily pollen. Lily pollen grains were incubated in pollen growth media containing 0–10 μg/mL poacic acid for 90 min at 22 °C. Although 97.7 ± 1.2% of pollen grains germinated after incubation in media without poacic acid, the pollen germination rate significantly decreased with increasing concentrations of poacic acid (Figure 1A–E); 93.5 ± 3.7%, 92.0 ± 5.7%, 56.2 ± 22.7%, or 17.2 ± 3.6% of pollen grains germinated in media containing 1, 2, 5, or 10 μg/mL poacic acid, respectively (Figure 1E). We used fluorescein diacetate staining [18] to assess the viability of pollen grains incubated in media containing poacic acid. More than 70% of the pollen was viable after incubation in media with or without 5 μg/mL poacic acid for 30 min (Figure 1F), indicating that the observed pollen germination defect in media containing poacic acid was not due to cell death.

Poacic acid also inhibited pollen tube growth in our study. When pollen grains were incubated in media without poacic acid for 90 min, the average pollen tube length was 52.5 ± 24.5 μm. In contrast, the average pollen tube length was 38.5 ± 14.1 μm, 28.7 ± 8.6 μm, 9.7 ± 5.6 μm, or 3.4 ± 6.8 μm when pollen grains were incubated in media containing 1, 2, 5, or 10 μg/mL poacic acid, respectively (Figure 1A–D,G). In addition to growth inhibition, the poacic acid treatment caused aberrant morphologies in pollen tubes. In contrast to the straight tube morphology observed in the medium without poacic acid (Figure 1H), some pollen tubes incubated in media containing 5 μg/mL poacic acid exhibited wavy morphologies (Figure 1I). We also observed pollen grains with two pollen tubes (Figure 1J).

Next, we examined the effect of poacic acid on pollen tube growth. Growing pollen tubes were transferred to solid pollen growth medium prepared on a microscope slide and observed under a differential interference contrast microscope. When the pollen tubes were transferred to media without poacic acid, they continued to grow (Figure 2A–C). The average length of the pollen tube during the 20 min observation period was 74 ± 63.8 μm (Figure 2G). In contrast, pollen tubes immediately stopped growing after transfer to media containing 5 μg/mL poacic acid (Figure 2D–G).

### 2.2. Poacic Acid Does Not Inhibit Callose Synthesis in Pollen Tubes

Callose (β-1,3-glucan) is a major component of the pollen cell wall. We examined whether poacic acid inhibited callose synthesis in pollen tubes. Growing pollen tubes were transferred to media containing 5 μg/mL poacic acid and incubated further. To visualize callose, pollen tubes were fixed and subjected to aniline blue staining. A weak aniline blue staining signal was observed in pollen tubes treated with poacic acid for 0 min (Figure 3A). We observed an increase in the aniline blue staining signal in pollen tubes incubated with poacic acid, indicating callose synthesis (Figure 3B,C). Continued callose synthesis was observed in both the apex and shank of the pollen tubes. The intensity of aniline blue staining at 10 μm (apex) and 80 μm (shank) from the apex increased with time (Figure 3D,E). No increase in aniline blue staining was observed in pollen tubes incubated with DMSO, unlike in those incubated with poacic acid (Appendix A). These results indicate that poacic acid did not inhibit callose synthesis in the pollen tubes.

### 2.3. Poacic Acid Inhibits Endocytosis and Secretion and Disrupts Actin Filaments in Pollen Tubes

Rapid pollen tube growth requires tip-focused cell expansion via vigorous exocytosis. Active retrograde endocytosis counterbalances exocytosis at the tip, which is crucial for pollen tube growth [19]. Therefore, we examined whether poacic acid affects membrane trafficking events. Confocal images of pollen tubes incubated with FM4-64, an endocytosis marker, for 30 min showed distinct peripheral plasma membrane staining and V-shaped apical staining (Figure 4A–C), as previously described [20]. In pollen tubes treated with 5 μg/mL poacic acid, only the plasma membrane signal was observed. We did not observe V-shaped apical staining, indicating the inhibition of endocytosis (Figure 4E). Using the autofluorescence of poacic acid, we analyzed the localization of poacic acid in the pollen tubes. Punctate signals of poacic acid autofluorescence were observed inside the pollen tube but not in the cell wall (Figure 4D), indicating that the added poacic acid was incorporated into the pollen tubes.

Next, we examined the effect of poacic acid on exocytosis. We used acid phosphatase, which is secreted from lily pollen tubes via the secretory pathway [21], as an exocytosis marker. The acid phosphatase activity in the medium of the pollen tube suspension increased over time without the addition of poacic acid. However, the acid phosphatase activity in the medium remained low when poacic acid was added at 5 μg/mL (Figure 4G), suggesting that poacic acid inhibits exocytosis.

The actin cytoskeleton is another cellular component required for rapid growth of pollen tubes. Actin filaments generate the driving force for cytoplasmic streaming, which is required for the long-distance transport of organelles and secretory vesicles [22,23]. We analyzed the effects of poacic acid on the actin filaments. Rhodamine-phalloidin staining of lily pollen tubes showed shank-localized actin filaments forming longitudinal bundles and short actin filaments in the subapical region (Figure 5A) [23]. The actin cytoskeleton in the subapical region of pollen tubes is known as an actin fringe [24,25]. In contrast to the long actin cables in the shank, the actin fringe consists of short and longitudinally oriented actin cables. The actin fringe at the subapical region disappeared when pollen tubes were incubated with 5 μg/mL poacic acid, but the actin cables in the shank remained (Figure 5B). When pollen tubes were treated with 100 μg/mL poacic acid, actin cable staining was lost (Figure 5C). These results suggest disorganization of actin structures in pollen tubes incubated in media containing poacic acid.

Next, we analyzed cytoplasmic streaming in lily pollen tubes in the presence or absence of poacic acid. The cytoplasmic streaming speed in pollen tubes was 1.09 ± 0.71 μm/s without poacic acid. When the medium was supplemented with 5 μg/mL of poacic acid, the cytoplasmic streaming speed decreased to 0.69 ± 0.35 μm/s (Figure 5D). However, we did not observe complete inhibition of the cytoplasmic streaming. The inhibition of pollen tube growth by poacic acid may not be due to the inhibition of cytoplasmic streaming.

The actin fringe is proposed to function in the accumulation of secretory vesicles in the tube tip region and to prevent organelle entry into the tip of the pollen tube [23,26,27,28]. We examined the effect of poacic acid on the distribution of lipid droplets, which contribute to pollen tube growth by providing triacylglycerols as an energy source [29]. The distribution of lipid droplets in pollen tubes depends on the actin cytoskeleton [30]. Nile Red staining revealed that lipid droplets were not present in the tip region of lily pollen tubes (Figure 5E). In contrast, lipid droplets were found in the tip region of pollen tubes less than 5 μm from the apex after incubation of pollen tubes in media containing 5 μg/mL poacic acid (Figure 5F,G). Transmission electron microscopy also revealed that the number of large lipid droplets (≥500 nm in diameter) increased in the tip region of pollen tubes incubated with poacic acid (Figure 5H,I). The observed misplacement of large organelles in the tip region was probably due to the actin fringe defect in poacic acid-treated pollen tubes.

### 2.4. Poacic Acid Affects the Growth of Arabidopsis thaliana Pollen Tubes and Roots

We also analyzed the effect of poacic acid on the growth of *Arabidopsis* pollen tubes. Since the efficiency of *Arabidopsis* pollen germination was not high under our in vitro culture conditions, we used semi in vitro pollen tube growth experiments. Pollen tubes that emerged from the cut end of the style were allowed to grow on pollen growth medium containing poacic acid. *Arabidopsis* pollen tubes grew on media containing 10 μg/mL poacic acid (Figure 6C), although significant growth inhibition was observed in lily pollen tubes at this concentration (Figure 1D,E,G). However, we observed swelling of the pollen tubes at the tip (Figure 6D). Pollen tubes grown on media containing 100 μg/mL poacic acid were short (Figure 6E). We observed a burst of pollen tubes at the tips (Figure 6F). These results indicate that poacic acid affects the growth of *Arabidopsis* pollen tubes. However, the effect was different from that observed in lily pollen tubes, and the effect was visible at a higher poacic acid concentration.

Poacic acid also affects the growth of *Arabidopsis* roots. The length of roots grown on media containing 100 μg/mL poacic acid was shorter than that on media without poacic acid (Figure 6G–I). Such growth inhibition was not observed in media containing 10 μg/mL poacic acid. These results suggest that the effects of poacic acid on plant cells are not limited to lily pollen tubes.

## 3. Discussion

Poacic acid is a novel natural antifungal agent identified in lignocellulosic hydrolysates from corn stover, which inhibits the growth of *S. cerevisiae* by inhibiting the synthesis of β-1,3-glucan [1,2]. Here, we showed that poacic acid also inhibited lily pollen germination and pollen tube growth. Poacic acid inhibited pollen tube growth at a concentration of 5 μg/mL, which was considerably lower than that required for yeast cell growth inhibition (IC50 = 111 μg/mL). This indicates that poacic acid has potent inhibitory activity in lily pollen tubes.

We found that the inhibition of pollen tube growth by poacic acid is not due to the inhibition of β-1,3-glucan (callose) synthesis. An increase in callose content was observed in the poacic acid-treated pollen tubes, indicating that callose synthesis was not inhibited (Figure 3). This was probably because the concentration of poacic acid (5 μg/mL) used to inhibit pollen tube growth was six-fold lower than the IC50 value of poacic acid for inhibiting β-1,3-glucan synthesis in budding yeast (31 μg/mL) [1]. Poacic acid directly binds to β-1,3-glucan in the yeast cell wall to inhibit β-1,3-glucan synthesis [1,2]. However, we did not observe poacic acid autofluorescence in the cell walls of poacic acid-treated pollen tubes (Figure 4D), probably because the amount of callose in the pollen tube cell wall may not be sufficient for poacic acid binding. The observed increase in callose content in poacic acid-treated pollen tubes (Figure 3) was probably due to continued callose synthesis in growth-arrested pollen tubes. A similar increase in callose content was observed in growth-arrested pollen tubes treated with phenylboronic acid, a boron antagonist [31]. It should be noted that callose synthesis is not essential for the growth of pollen tubes. The *Arabidopsis cals5-3* mutant is defective in callose synthesis in pollen tubes, but the mutant pollen tubes grow and retain their fertility [32].

Our results indicate that poacic acid strongly inhibits both secretion and endocytosis in the pollen tubes (Figure 4). The rapid growth of pollen tubes requires expansion of the plasma membrane at the apex via secretion [33,34]. Sustained pollen tube growth also requires membrane recycling via endocytosis [35]. The inhibition of these apical membrane trafficking processes by poacic acid likely resulted in the observed arrest of pollen tube growth. The actin fringe structure was disrupted in pollen tubes treated with poacic acid, but actin cables were retained in the shank region (Figure 5B). The actin fringe physically prevents the entry of other organelles into the apical region, either directly or by accumulating secretory vesicles in the subapical region [27]. Consistent with the disappearance of the actin fringe, we observed lipid droplet penetration in the subapical region (Figure 5F,G).

It remains unclear how poacic acid inhibits membrane traffic and the organization of the actin cytoskeleton in pollen tubes. As the autofluorescence of poacic acid was observed in pollen tubes, the target of poacic acid in pollen tubes is most likely located inside these cells. Actin cables that align along the elongation axis of the pollen tube in the shank region play essential roles in cytoplasmic streaming within the pollen tube, which delivers organelles, including secretory vesicles, to the apical region of the pollen tube [22,26,27]. The actin fringe in the subapical region is a bundle of longitudinally oriented short actin filaments that plays a crucial role in targeting secretory vesicles to the pollen tube apex and endocytosis [23,28]. It is possible that the disruption of the actin fringe by poacic acid caused inhibition of membrane traffic, which in turn resulted in inhibition of pollen tube growth. Actin cables and cytoplasmic streaming were still observed in the 5 μg/mL poacic acid-treated pollen tubes (Figure 5B,D). Actin cables disappeared in pollen tubes treated with 100 μg/mL poacic acid (Figure 5C). It is likely that actin fringes are more sensitive to drugs, including poacic acid, than actin cables. Consistent with our observations, treatment of lily pollen tubes with low concentrations of latrunculin B, which inhibits actin polymerization, caused the disappearance of the actin fringe but the retention of arrays of actin cables in the shank [36]. However, it remains unclear whether poacic acid directly inhibits actin polymerization. The actin fringe also disappears due to changes in physiological conditions in the pollen tubes, such as the Ca^2+^ concentration and pH gradient [23]. Increased reactive oxygen species levels inhibit actin polymerization by altering the Ca^2+^ gradient within the pollen tube [37].

It is also possible that poacic acid directly inhibits membrane trafficking. We observed punctate autofluorescence of poacic acid in pollen tubes, suggesting the binding of poacic acid to membrane structures (Figure 4D), which may result in the inhibition of endocytosis and exocytosis. The disruption of the actin fringe has been reported in pollen tubes, where membrane trafficking is compromised. Pollen tubes of *Arabidopsis vps9* mutant that are defective in the function of guanine nucleotide exchange factor for Rab5 family GTPases, ARA7 and RHA1, showed defects in the endosome-vacuole transport and also showed defects in the actin fringe organization [38,39]. Pollen tubes of *Picea meyeri* treated with brefeldin A showed defects in actin fringe organization in the subapical region [14]. The disappearance of the actin fringe in poacic acid-treated lily pollen tubes was probably the result of defects in membrane trafficking. Although the inhibitory mechanism of poacic acid on lily pollen tube growth remains unclear, our study indicates that poacic acid affects lily pollen tube growth differently than it affects budding yeast.

Poacic acid also affected the growth of *Arabidopsis* pollen tubes. However, growth inhibition of *Arabidopsis* pollen tubes was not evident in media containing 10 μg/mL poacic acid, which significantly inhibited the growth of lily pollen tubes. Pollen tube bursting was observed when *Arabidopsis* pollen tubes were grown in media containing 100 μg/mL poacic acid (Figure 6E,F). This suggests that poacic acid may also affect the apical region of *Arabidopsis* pollen tubes. It is unclear whether poacic acid affects the actin cytoskeleton in *Arabidopsis* pollen tubes. Notably, pollen tubes of *Arabidopsis* mutants RabA4, which is involved in vesicle secretion, exhibit tube tip swelling, and mutants of ARMADILLO REPEAT ONLY1, which is required for actin organization in pollen tubes, exhibit tube tip swelling and bursting [40,41]. Poacic acid also inhibited *Arabidopsis* root growth, suggesting that the effects of poacic acid are not limited to the pollen tubes. Root growth inhibition was observed in media containing 100 μg/mL poacic acid. The sensitivity of *Arabidopsis* cells to poacic acid may be lower than that of lily cells.

Our study revealed that poacic acid has inhibitory effects on plant cells, despite being derived from the plant cell wall. The primary target of poacic acid is likely located inside the plant cell, which differs from that in fungal cells. It remains unclear whether poacic acid exists in the plant cell wall and regulates physiological processes in plant cells. Poacic acid inhibited the activities of some plant tissues at a concentration exhibiting full antifungal activity. Poacic acid may also act on the plant cells of various angiosperm species, including monocots and dicots, at concentrations used as a pesticide. Although poacic acid is a potential pesticide candidate, further studies on its effects on plant cells are needed, especially regarding its application to living plants.

## 4. Materials and Methods

### 4.1. Plant Materials and Growth Conditions and Chemical Material

*Lilium longiflorum* flowers were purchased from local flower shops. Anthers containing dehydrated pollen were collected and stored at −30 °C until use. *Arabidopsis thaliana* ecotype Columbia seeds were surface sterilized and grown on soil or Murashige and Skoog (MS) medium (Fuji Film Wako, Osaka, Japan) containing 0.7% agar at 22 °C under continuous light. Poacic acid [1] is a gift from Fachuang Lu, Ruili Gao, Jeff Piotrowski, and John Ralph at the University of Wisconsin, and was dissolved in dimethyl sulfoxide (DMSO) at 100 μg/mL for the stock solution.

### 4.2. Analysis of Pollen Growth

Pollen growth medium used in this study is 1.5 mM Ca(NO_3_)_2_, 0.01% (*w*/*v*) boric acid, 5% (*w*/*v*) sucrose, 15 mM 2-(N-morpholino)ethanesulfonic acid (MES)-KOH, pH 5.6. For the solid medium, low-gelling-temperature agarose (NuSieve GTG, Takara Bio Inc., Shiga, Japan) was added at a concentration of 1.5% (*w*/*v*). Lily pollen grains from one anther were suspended in 5 mL of pollen growth medium and transferred to a 60 mm Petri dish or 800 μL was dispensed into each well of 24 well plates. Pollen culture was performed at 22 °C with gentle agitation. The medium without poacic acid contained DMSO at the same concentration as that in the medium containing poacic acid.

To analyze the pollen germination rate, pollen tube length, and pollen tube morphology, the pollen grains were incubated for 90 min in a pollen growth medium containing poacic acid. Pollen tubes and grains were fixed with a 3:1 (*v*/*v*) mixture of ethanol/acetic acid for 5 min and washed three times with phosphate-buffered saline (PBS).

To analyze pollen tube growth, pollen tubes were transferred to a solid pollen growth medium containing poacic acid on a glass slide and incubated in a humidified chamber. Pollen tubes were observed and photographed after incubation for 0, 10, and 20 min at 22 °C.

To investigate the effect of poacic acid on growing pollen tubes, 200 μL of pollen growth medium containing poacic acid at a concentration fivefold the final concentration was added to 800 μL of pollen suspension and incubated for 30 min. To analyze cytoplasmic streaming, time-lapse images of the pollen tubes were acquired at 1 s intervals for 1 min. To analyze pollen viability, pollen grains and pollen tubes from 1 mL of pollen culture suspension were incubated in 1 mL of 7% (*w*/*v*) sucrose and 0.02% (*w*/*v*) fluorescein diacetate for 30 min at room temperature in the dark.

Semi in vitro *Arabidopsis* pollen growth experiments were performed as described by [42], with modifications. Pistils pollinated with pollen were cut with a knife at the junction between the style and ovary and placed horizontally on 0.01% boric acid (*w*/*v*), 5 mM CaCl_2_, 5 mM KCl, 1 mM MgSO_4_, 10% sucrose (*w*/*v*), pH 7.5, and 1.5% NuSieve GTG agarose. In the case of media containing poacic acid, the media were adjusted to pH 7.5 using NaOH solution. The samples were incubated for 7–8 h at 22 °C in a humidity chamber.

Pollen grains and pollen tubes were observed under a Zeiss Stemi 305 stereo microscope (Carl Zeiss, Oberkochen, Germany) or a BX51 microscope (Olympus, Tokyo, Japan) equipped with a DP70 cooled CCD camera (Olympus, Tokyo, Japan). Fluorescein fluorescence images were captured using a BX51 microscope (Olympus, Tokyo, Japan) with a fluorescein filter set. MetaMorph (Version 6.3r3, Universal Imaging, Brandywine, PA, USA) was used for image acquisition.

### 4.3. Pollen Staining

For aniline blue staining, pollen tubes were fixed with a 3:1 (*v*/*v*) mixture of ethanol/acetic acid for 5 min, washed three times with water, and suspended in 0.1% (*w*/*v*) decolorized aniline blue and 0.1 M potassium phosphate (pH 11).

For actin staining, pollen tubes were fixed in 4% (*w*/*v*) paraformaldehyde, 10% (*w*/*v*) sucrose, and 50 mM piperazine-1,4-bis(2-ethanesulfonic acid) (PIPES)-KOH, pH 6.9, under vacuum for 5 min and further fixed for 1 h. Fixed samples were washed three times with 50 mM PIPES-KOH, pH 6.9, and incubated with 60 nM rhodamine X-conjugated phalloidin (FUJIFILM Wako, Tokyo, Japan), 10% (*w*/*v*) sucrose, 1% (*v*/*v*) DMSO, 0.01% (*v*/*v*) Nonidet P-40, and 50 mM PIPES-KOH, pH 6.9 for 1 h in the dark.

For Nile Red staining, pollen tubes were fixed with 4% formaldehyde, 0.5% glutaraldehyde, 5 mM MgSO_4_, 5 mM EGTA, 0.1 M mannitol, and 50 mM PIPES-KOH (pH 6.9) under vacuum for 1 h, and further fixed in the same solution overnight at room temperature. The fixed pollen tubes were washed three times and then incubated in 0.4 μg/mL Nile Red (FUJIFILM Wako, Tokyo, Japan), 5 mM MgSO_4_, 5 mM EGTA, 0.1 M mannitol, and 50 mM PIPES-KOH, pH 6.9 for 5 min. Pollen tube staining with FM4-64 dye was performed according to [20].

The samples were observed using a Leica TCS-SP8 confocal laser scanning microscope (Leica Microsystems, Wetzlar, Germany) with a 20× water-immersion objective lens (PL APO CS2 20×/0.75 W CORR HC; Leica Microsystems), 40× water-immersion objective lens (PL APO CS2 40×/1.10 W CORR HCX; Leica Microsystems), or 60 × water-immersion objective lens (PL APO CS2 60×/1.10 W CORR HCX; Leica Microsystems). Images of aniline blue fluorescence were captured at 410–471 nm after excitation at 405 nm using a diode laser. Rhodamine X and Nile Red fluorescence images were captured at 557–650 nm after excitation at 552 nm using a solid-state laser. FM4-64 fluorescence images were captured at 650–785 nm after excitation at 488 nm using a solid-state laser. Images of poacic acid autofluorescence were captured at 517–552 nm after excitation at 405 nm with a diode laser.

### 4.4. Acid Phosphatase Assay

Pollen grains from one anther were washed five times with a pollen growth medium to remove acid phosphatase bound to the pollen surfaces. Pollen grains were suspended in 12 mL of pollen growth medium and incubated at room temperature for 90 min in a 60 mm Petri dish with gentle agitation. The pollen suspension was divided into two parts. Each culture was mixed with an equal volume of pollen growth medium containing either 10 μg/mL poacic acid or no poacic acid. A 200 μL of pollen culture supernatant was collected and immediately frozen at −20 °C. Acid phosphatase activity was measured according to [43] within three days of collection. All assays were performed in triplicates.

### 4.5. Electron Microscopy

Pollen tubes were fixed in 2.5% (*v*/*v*) glutaraldehyde, 5% (*w*/*v*) sucrose, and 100 mM phosphate buffer (pH 7.2) for 3 h at 4 °C. The fixed samples were washed with 100 mM phosphate buffer (pH 7.2) and further fixed with 2% OsO_4_ for 2 h. The samples were dehydrated using an ethanol series, infiltrated, and embedded in Quetol-651. Ultrathin sections (80 nm) were stained with aqueous uranyl acetate and lead citrate and observed using a Hitachi H-7650 electron microscope (Hitachi High-Tech, Tokyo, Japan) at 80 kV.

### 4.6. Quantification and Statistical Analysis

Pollen tube length, velocities of cytoplasmic streaming in pollen tubes, size of regions lacking Nile Red-stained vesicles in pollen tubes, and lipid droplet size were measured using the Fiji software (ver. 2.16.0/1.54p) [44]. The velocity of cytoplasmic streaming was calculated based on the distance traveled by the cytoplasmic vesicles over 10 s. Aniline blue fluorescence intensity was measured using Fiji. Fluorescence at each position was measured in a straight line perpendicular to the pollen tube, and the maximum intensity was defined as the fluorescence intensity. Experiments involving quantitative analyses were conducted using at least three biological replicates. The effects of the treatments were analyzed using one-way analysis of variance (ANOVA). Tukey’s test was used to compare the significance of the differences among the treatments. A two-sided Student’s *t*-test was used to analyze the significant differences between the two groups.

## Figures and Tables

**Figure 1 plants-14-03093-f001:**
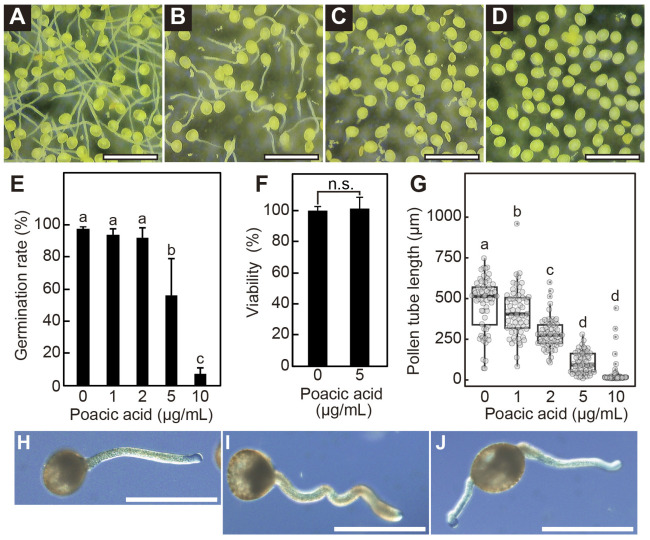
Poacic acid inhibited the germination of lily pollen and the growth of pollen tubes. Lily pollen was incubated in pollen growth medium containing various concentrations of poacic acid for 90 min. Images of pollen incubated in media containing (**A**) 0 μg/mL, (**B**) 2 μg/mL, (**C**) 5 μg/mL, and (**D**) 10 μg/mL of poacic acid are shown. Scale bars = 0.5 mm. (**E**) Pollen germination rates in the presence of various poacic acid concentrations. The pollen germination rate was calculated from more than 200 pollen grains for each culture condition. The averages of at least three independent experiments are shown with standard deviations. Different letters indicate significant differences according to Tukey’s test (*p* < 0.05). (**F**) Pollen viability after incubation in media with or without 5 μg/mL poacic acid for 30 min. Relative pollen viability is shown with the viability of pollen incubated in media containing 0 μg/mL poacic acid, set at 100%. Relative viability was calculated from over 200 pollen grains for each culture condition. The averages of at least three independent experiments are shown with standard deviations. Statistical differences were calculated using Student’s *t*-test. n.s., not significant. (**G**) Pollen tube length after incubation in media containing various concentrations of poacic acid for 90 min. The lengths of at least 50 pollen tubes were measured for each concentration. Different letters indicate significant differences according to Tukey’s test (*p* < 0.05). Images of pollen tubes incubated in media containing (**H**) 0 μg/mL or (**I**,**J**) 5 μg/mL poacic acid for 90 min are shown. Scale bars = 100 μm.

**Figure 2 plants-14-03093-f002:**
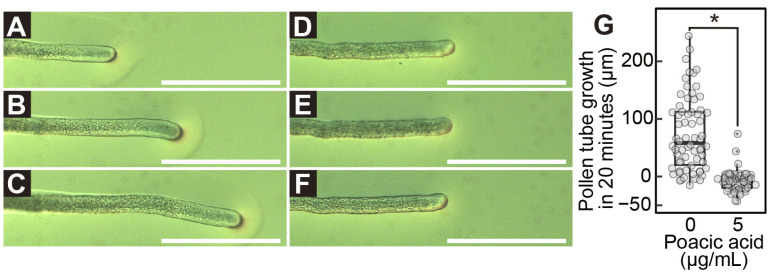
Immediate growth arrest of pollen tubes by poacic acid. Growing pollen tubes were transferred to solid media containing 0 μg/mL (**A**–**C**) or 5 μg/mL (**D**–**F**) of poacic acid and incubated at 22 °C. Microscopic images of pollen tubes at 0 min (**A**,**D**), 10 min (**B**,**E**), and 20 min (**C**,**F**) after transfer are shown. Scale bars = 100 μm. (**G**) Length of pollen tube growth during 20 min incubation in media containing 0 μg/mL or 5 μg/mL of poacic acid. At least 60 pollen tubes were measured under each condition. Statistical differences were calculated using Student’s *t*-test. The asterisk indicates *p* < 0.01.

**Figure 3 plants-14-03093-f003:**
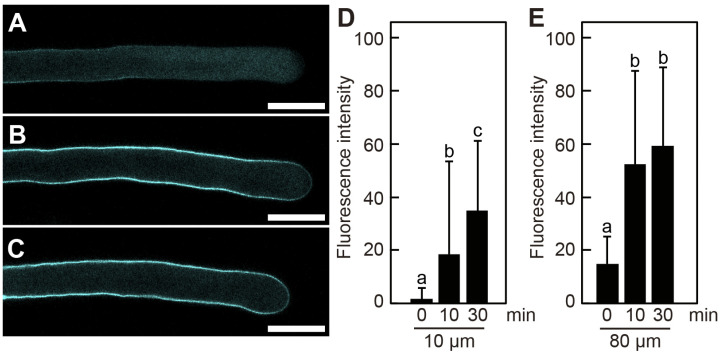
Callose synthesis in pollen tubes in media containing poacic acid. Growing pollen tubes were transferred to media containing 5 μg/mL poacic acid and incubated at 22 °C. Pollen tubes were fixed, stained with aniline blue, and analyzed using a confocal laser microscope. (**A**–**C**) Confocal images of pollen tubes incubated for 0 min (**A**), 10 min (**B**), or 30 min (**C**) in media containing 5 μg/mL poacic acid. Scale bars = 30 μm. Intensities of aniline blue fluorescence at (**D**) 10 μm and (**E**) 80 μm from the pollen tube apex. Fluorescence intensities of at least 20 pollen tubes were measured. Different letters indicate significant differences according to Tukey’s test (*p* < 0.05).

**Figure 4 plants-14-03093-f004:**
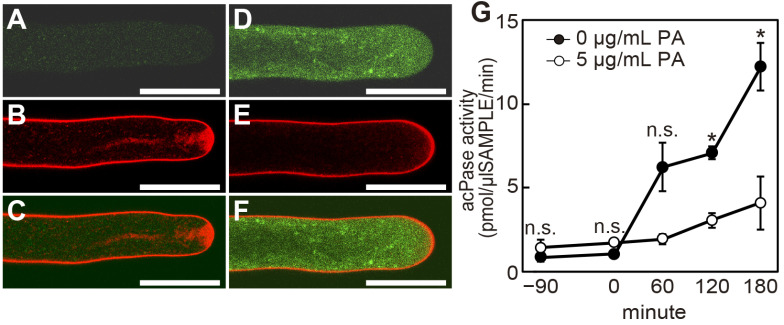
Poacic acid affects membrane trafficking in pollen tubes. Growing pollen tubes that were incubated in media containing 0 μg/mL (**A**–**C**) or 5 μg/mL (**D**–**F**) poacic acid at 22 °C for 30 min were further incubated with 5 μM FM4-64 for 30 min and observed under a confocal laser scanning microscope. Images of poacic acid autofluorescence (**A**,**D**), FM4-64 fluorescence (**B**,**E**), and merged images (**C**,**F**) are shown in the figure. Scale bars = 30 μm. (**G**) Pollen tubes that had grown in pollen growth medium for 90 min were incubated with 0 μg/mL or 5 μg/mL poacic acid at 22 °C. The acid phosphatase activity in the culture supernatant was measured at each time point. The time of poacic acid addition was set to 0 min. The average acid phosphatase activities of at least three independent experiments are shown with standard deviations. Statistical differences were calculated using Student’s *t*-test. The asterisk indicates *p* < 0.01. n.s., not significant.

**Figure 5 plants-14-03093-f005:**
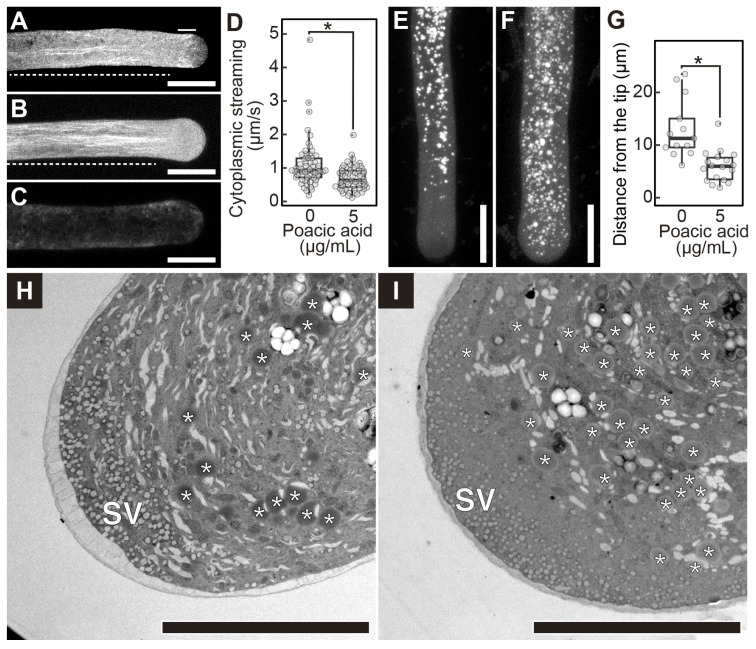
Poacic acid disrupts actin filaments in pollen tubes. Growing pollen tubes were transferred to media containing 0 μg/mL (**A**), 5 μg/mL (**B**), or 100 μg/mL (**C**) poacic acid and incubated for 30 min at 22 °C. Pollen tubes were fixed, stained with rhodamine-phalloidin, and observed under a confocal laser scanning microscope. Images of rhodamine-phalloidin fluorescence are shown in the figure. The solid line indicates the subapical region with the actin fringes. Dotted lines indicate the pollen shank region containing actin cables. Scale bars = 20 μm. (**D**) Velocities of cytoplasmic streaming in pollen tubes in media containing 0 μg/mL or 5 μg/mL of poacic acid. The velocities of cytoplasmic streaming were calculated from the movement of vesicles in the pollen tubes using time-lapse images of the pollen tubes (*n* = 50). Statistical differences were calculated using Student’s *t*-test. The asterisk indicates *p* < 0.01. (**E**,**F**) Images of Nile Red-stained pollen tubes incubated in pollen growth media containing 0 μg/mL (**E**) or 5 μg/mL (**F**) of poacic acid. (**G**) Size of regions without Nile Red-stained vesicles in pollen tubes incubated in media containing 0 μg/mL or 5 μg/mL of poacic acid. The lengths of the regions from the apex of the pollen tubes were measured. At least 13 pollen tubes were used for each condition in each experiment. Statistical differences were calculated using Student’s *t*-test. The asterisk indicates *p* < 0.01. (**H**,**I**) Transmission electron micrographs of pollen tubes incubated in media containing 0 μg/mL (**H**) or 5 μg/mL (**I**) poacic acid. Asterisks indicate lipid droplets ≥ 500 nm in diameter. SV, secretory vesicles. Scale bars = 10 μm.

**Figure 6 plants-14-03093-f006:**
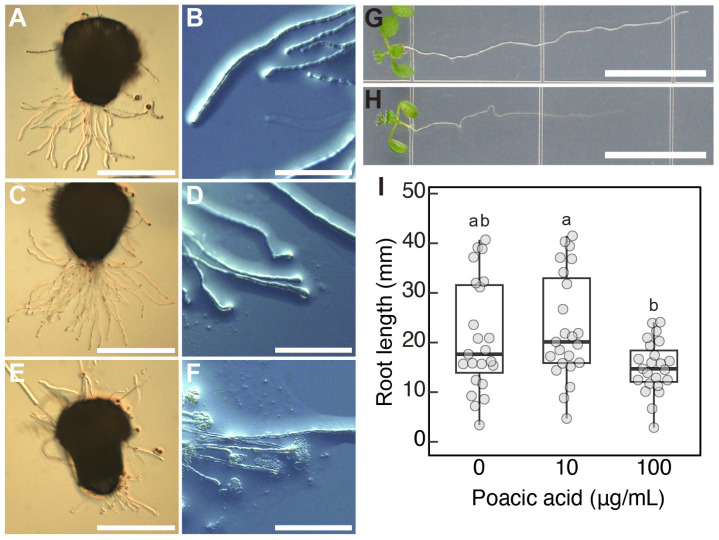
Effects of poacic acid on the pollen tubes and roots of *Arabidopsis thaliana*. (**A**–**F**) Images of pollen tubes germinated on excised stigma and grown on media containing 0 μg/mL (**A**,**B**), 10 μg/mL (**C**,**D**), or 100 μg/mL (**E**,**F**) poacic acid. (**A**,**C**,**E**) Images of excised pistils and pollen tubes. (**B**,**D**,**F**) Images of pollen tubes grown on media. (**G**,**H**) Images of roots grown for 7 days on media containing 0 μg/mL (**G**) or 100 μg/mL (**H**) poacic acid are shown. (**I**) Root length of plants grown on media containing 0 μg/mL, 10 μg/mL, or 100 μg/mL of poacic acid for 7 days. At least 20 roots were measured for each condition in each experiment. Different letters are significantly different from each other according to Tukey’s test (*p* < 0.05). Scale bars = 500 μm (**A**,**C**,**E**), 100 μm (**B**,**D**,**F**), 10 mm (**G**,**H**).

## Data Availability

The original contributions presented in this study are included in the article/Appendix A. Further inquiries can be directed to the corresponding author.

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
