# Peer review of "A Plant-Derived Antifungal Agent, Poacic Acid, Inhibits Germination and Tube Growth of Lily Pollen"

_plants, 2025, doi:10.3390/plants14193093_

Round 1
Reviewer 1 Report (New Reviewer)
Comments and Suggestions for Authors
This is a small but interesting story, well written and mostly (not entirely - see below) well supported by experimental evidence.
The only factual concern I have is related to the data presented in Figure 3, where I am missing images and measurements from mock-treated pollen tubes (i.e. identical conditions but DMSO instead of the poacic acid solutions). Also, in Figure 3D quantitative (statistical) comparisons between mock-treated and poacic acid-treated pollen have to be shown for each timepoint (instead of the current between-timepoint comparisons), in order to assess the extent of inhibition of callose synthesis (or document the presumed lack of inhibition).
There are some issues related to the statistical procedures used, but the differences shown are quite dramatic and I am nearly sure they will be significant even when evaluated using appropriate methods. Specifically, there are some cases (Fig. 1G, Fig. 3D, Fig. 6I) when t-test was used to perform multiple comparisons but apparently without any multiplicity correction. ANOVA would be a more appropriate method here. Also, it is not clear how was statistical significance evaluated in Fig. 1E and Fig. 1F. Since these are categorical data, Chi square (with multiplicity correction) would be an appropriate method (but in that case a representative experiment should be shown without error bars), Or are these averages from N = 3 independent replicates, with significance evaluated by ANOVA in E and t-test in F? Please clarify.
There are a couple of minor, mostly formal aditional issues that need to be addressed:
- Why are random bits of text in red or dark green? Please unify text color to black.
- There is no need to repeat numeric quantitative data in the text if they are already shown in graphs (relevant throughout the paper).
- L. 178-179: This is a bit of overinterpretation, streaming inhibition may still be partially contributing.
- L. 258-259: This statement (also present in some form in the Abstract) is not supported by experimental data - see previous comment on Fig. 3.
- L. 301 - typo ("poaic" instead of "poacic")
- L. 323-324: However, experimental conditions were incomparable (different media), thus this observation cannot be interpreted as evidence for a different mechanism.
- Part 4.6: How was velocity of cytoplasmic streaming measured?
Author Response
Thank you for your time and consideration. Please find our detailed responses below and the corresponding revisions in the re-submitted files. The changes made in the revised manuscript are marked in red.
Comment 1: This is a small but interesting story, well written and mostly (not entirely - see below) well supported by experimental evidence.The only factual concern I have is related to the data presented in Figure 3, where I am missing images and measurements from mock-treated pollen tubes (i.e. identical conditions but DMSO instead of the poacic acid solutions). Also, in Figure 3D quantitative (statistical) comparisons between mock-treated and poacic acid-treated pollen have to be shown for each timepoint (instead of the current between-timepoint comparisons), in order to assess the extent of inhibition of callose synthesis (or document the presumed lack of inhibition).
Response 1: Thank you for your valuable comment. We agree with your suggestion that it is essential to compare mock-treated and poacic acid-treated pollen tubes to demonstrate the effect on callose synthesis. To address this, we analyzed pollen tubes incubated with or without poacic acid for 30 min using aniline blue staining. We did not observe increased callose synthesis in pollen tubes incubated without poacic acid. The results are presented in Figure S1. The results are presented in Section 2.2. (lines 135–136).
Comment 2: There are some issues related to the statistical procedures used, but the differences shown are quite dramatic and I am nearly sure they will be significant even when evaluated using appropriate methods. Specifically, there are some cases (Fig. 1G, Fig. 3D, Fig. 6I) when t-test was used to perform multiple comparisons but apparently without any multiplicity correction. ANOVA would be a more appropriate method here. Also, it is not clear how was statistical significance evaluated in Fig. 1E and Fig. 1F. Since these are categorical data, Chi square (with multiplicity correction) would be an appropriate method (but in that case a representative experiment should be shown without error bars), Or are these averages from N = 3 independent replicates, with significance evaluated by ANOVA in E and t-test in F? Please clarify.
Response 2: We appreciate your careful reading of and your valuable feedback on this manuscript. We performed statistical analyses using one-way analysis of variance (ANOVA) and Tukey's test on the results shown in Figure 1E and F, 3D, and 6I. We have added the methods for ANOVA and Tukey's test to Section 4.6 of the revised manuscript. (lines 445–447). We also revised the legend for Figure 1 to clarify that Figures 1E and 1F show the mean and standard deviation of three independent experiments (lines 97–108).
Comment 3: There are a couple of minor, mostly formal aditional issues that need to be addressed: Why are random bits of text in red or dark green? Please unify text color to black.
Response 3: According to your comment, we have unified the text color to black for all instances.
Comment 4: There is no need to repeat numeric quantitative data in the text if they are already shown in graphs (relevant throughout the paper).
Response 4: Thank you for your valuable comment. Because the numerical quantitative data are not shown in the graphs, we have included them in the main text.
Comment 5: L. 178-179: This is a bit of overinterpretation; streaming inhibition may still be partially contributing.
Response 5: According to your comments, we have revised this sentence. The revised sentence is “The inhibition of pollen tube growth by poacic acid may not be due to the inhibition of cytoplasmic streaming.”. (lines 183–184).
Comment 6: L. 258-259: This statement (also present in some form in the Abstract) is not supported by experimental data - see previous comment on Fig. 3.
Response 6: We did not observe enhanced callose synthesis in pollen tubes incubated in media without adding poacic acid (Figure S1). We have retained this statement in the revised manuscript.
Comment 7: L. 301 - typo ("poaic" instead of "poacic")
Response 7: We corrected all instances of the “poaic acid” typo.
Comment 8: L. 323-324: However, experimental conditions were incomparable (different media), thus this observation cannot be interpreted as evidence for a different mechanism.
Response 8: We removed the phrase "but the effects may differ from those on lily pollen tubes.".
Comment 9: Part 4.6: How was velocity of cytoplasmic streaming measured?
Response 9: We have added the method for measuring the velocity of cytoplasmic streaming in Section 4.6 of the revised manuscript. (lines 440–441).
Reviewer 2 Report (New Reviewer)
Comments and Suggestions for Authors
This is a well-executed and scientifically sound study that provides important and novel insights. The authors convincingly demonstrate that poacic acid, a promising plant-derived antifungal compound, has significant and previously unknown effects on plant cell physiology, specifically in pollen tubes. The work is of high quality, the experiments are well-designed and appropriate, and the data robustly support the conclusions. The findings are highly relevant for the potential development of poacic acid as a pesticide, as they highlight a crucial caveat: its phytotoxicity.
I have the following comments to make.
Comment 1: The title is accurate but could be slightly more specific. It focuses on the "what" (inhibition) but not the novel "how" (non-fungal mechanism). A minor tweak could emphasize this, e.g., "...via disruption of membrane traffic and the actin cytoskeleton".
Comment 2: The discussion appropriately acknowledges the uncertainty around the precise primary target of poacic acid inside the plant cell. However, the two possibilities (direct action on actin vs. direct action on membranes) could be discussed with slightly more depth. The punctate autofluorescence pattern (Fig 4D) is intriguing evidence for direct membrane interaction. Could this be quantifying or described in more detail?
Comment 3: The figure panels are referred to as A-E, G, I, J in the results text, but the legend also mentions H. Figure 1H is not discussed in the results section (2.1). Please ensure all panels are described or remove the redundant reference.
Comment 4: The results show that 100 µg/mL poacic acid disrupts actin cables (Fig 5C) and inhibits Arabidopsis root growth (Fig 6I), but the yeast IC50 is 111 µg/mL. This could be briefly noted in the discussion to emphasize that at full antifungal concentrations, the phytotoxic effects on some plant tissues would be severe.
Comment 5: There are a few very minor grammatical errors and typos (e.g., "poaic acid" instead of "poacic acid" in section 4.2, "Ca^2-" instead of "Ca²⁺" in the discussion). These should be corrected in a final proofread.
Author Response
Thank you for your time and consideration. Please find our detailed responses below and the corresponding revisions in the re-submitted files. The changes made in the revised manuscript are marked in red.
Comment 1: The title is accurate but could be slightly more specific. It focuses on the "what" (inhibition) but not the novel "how" (non-fungal mechanism). A minor tweak could emphasize this, e.g., "...via disruption of membrane traffic and the actin cytoskeleton".
Response 1: We appreciate your valuable suggestion. However, it is difficult to draw a clear conclusion regarding the mechanism by which poacic acid inhibits pollen growth. Therefore, we will not change the title of the manuscript.
Comment 2: The discussion appropriately acknowledges the uncertainty around the precise primary target of poacic acid inside the plant cell. However, the two possibilities (direct action on actin vs. direct action on membranes) could be discussed with slightly more depth. The punctate autofluorescence pattern (Fig 4D) is intriguing evidence for direct membrane interaction. Could this be quantifying or described in more detail?
Response 2: The autofluorescence of poacic acid is weak, making the accurate quantification of its localization in membrane structures difficult. Currently, we lack appropriate markers for membranous structures in lily pollen tubes. It is difficult to provide further discussion regarding the target of poacic acid in this study.
Comment 3: The figure panels are referred to as A-E, G, I, J in the results text, but the legend also mentions H. Figure 1H is not discussed in the results section (2.1). Please ensure all panels are described or remove the redundant reference.
Response 3: According to your comments, we have added a description of Figure 1H in Section 2.1. (lines 88–91).
Comment 4: The results show that 100 μg/mL poacic acid disrupts actin cables (Fig 5C) and inhibits Arabidopsis root growth (Fig 6I), but the yeast IC50 is 111 μg/mL. This could be briefly noted in the discussion to emphasize that at full antifungal concentrations, the phytotoxic effects on some plant tissues would be severe.
Response 4: As recommended, we have added the sentence "Poacic acid inhibited the activities of some plant tissues at a concentration exhibiting full antifungal activity." to the Discussion (lines 338–340).
Comment 5: There are a few very minor grammatical errors and typos (e.g., "poaic acid" instead of "poacic acid" in section 4.2, "Ca^2-" instead of "Ca²⁺" in the discussion). These should be corrected in a final proofread.
Response 5: We corrected all instances of the “poaic acid” typo. We could not locate "Ca^2-" in the manuscript.
Round 2
Reviewer 1 Report (New Reviewer)
Comments and Suggestions for Authors
I am happy to recommend acceptance of this manuscript for publication, since all my concerns have been adequately addressed.
This manuscript is a resubmission of an earlier submission. The following is a list of the peer review reports and author responses from that submission.
Round 1
Reviewer 1 Report
Comments and Suggestions for Authors
This study reports for the first time the inhibitory effects of poacic acid—a plant-derived antifungal agent—on pollen germination and pollen tube growth in Lilium longiflorum, while documenting cellular phenotypes such as disorganization of the subapical actin fringe and disruption of membrane trafficking. Although the experimental design is methodologically sound and data presentation meets technical standards, the core scientific rationale suffers from a fundamental flaw: The observed phenotypes—growth arrest, cytoskeletal disorganization, and inhibition of endocytosis/exocytosis —represent common non-specific stress responses in pollen tubes, a system exquisitely sensitive to environmental perturbations (e.g., osmotic shifts, pH fluctuations, or solvent toxicity). Crucially, the conclusion contradicts the established antifungal mechanism of poacic acid, which targets β-1,3-glucan synthesis (Piotrowski et al., 2015), as the authors explicitly demonstrate no inhibition of callose deposition in pollen tubes. Consequently, the study fails to reveal novel biological mechanisms and offers negligible insights into poacic acid’s phytotoxicity or its implications for agrochemical development.
Reviewer 2 Report
Comments and Suggestions for Authors
The manuscript presents valuable data on how poacic acid, a plant-derived antifungal compound, impacts pollen germination and tube growth in lily and Arabidopsis. The experimental setup is appropriate, and the observations are clearly described. The topic is relevant to those studying plant cell biology and chemical effects on plant reproduction.
There are a few areas that could be improved before publication:
1. Please explain more clearly how poacic acid affects actin structure. The results suggest changes in the cytoskeleton, but the underlying cause remains unclear.
2. The methods section should include more detail about the statistical analysis used, such as the number of replicates and how significance was determined.
3. Some of the figures, especially microscopy images, would benefit from clearer labels or improved contrast to highlight the described structures.
4. As only two plant species were tested, it would be helpful to include a brief discussion about how the findings might apply to other species.
Comments on the Quality of English LanguageThe manuscript is generally understandable, but some sections would benefit from clearer phrasing and improved sentence structure.
Reviewer 3 Report
Comments and Suggestions for Authors
The article by Kobayashi et al. “plant-derived antifungal agent, poacic acid, inhibits germination and tube growth of lily pollen” is very interesting and very well written.
General
Please put Arabidopsis in italics throughout the manuscript
Introduction
Line 69 – I don’t understand the sentence “These results indicate that although poacic acid is a plant-derived compound, it affects plants’ cellular processes”.
In the way it is written, it looks like a conclusion. Many plant-derived compounds affect plant cellular processes such as colchicine, oryzalin, etc.
Results
Line 104 – You mentioned that ”Growing pollen tubes were transferred to a solid pollen growth medium prepared on a microscope slide”. But in the methods section (lines 337-338) you refer that you fixed the pollen tubes in ethanol acetic-acid, a fixative which helps to preserve tissue structure and morphology, and thus it kill cells. How can the tubes continue to grow?
Or those grains are not fixed?
In the methods you aggregate several procedures in subtopic “Microscopy”. It is better to clarify this.
Materials and Methods
Lines 327-228 – “In the sentence Lily pollen grains from one anther were suspended in 5 mL of pollen growth medium and incubated at 22°C with …” Where did you put the medium? In a petri dish?, Erlenmeyer?